# FLAIR: FEDERATED LEARNING WITH AUGMENTED AND IMPROVED FEATURE REPRESENTATIONS

## ABSTRACT

Federated Learning (FL) enables collaborative model training across decentralized clients while preserving data privacy. However, its performance declines in challenging heterogeneous data settings. To mitigate this, existing FL frameworks not only share locally trained parameters but also exchange additional information – such as control variates, client features, and classifier characteristics – to address the effects of class imbalance and missing classes. However, this leads to increased communication costs and heightened risks of privacy breaches. To strike a balance between communication efficiency, privacy protection, and adaptability to heterogeneous data distributions, we propose FLAIR, a novel FL approach with augmented and improved feature representations. FLAIR utilizes Class Variational Autoencoders (CVAE) for feature augmentation, mitigating class imbalance and missing class issues. It also incorporates Reptile meta-training to facilitate knowledge transfer between model updates, adapting to dynamic feature shifts. To generalize model update, FLAIR shares only local CVAE parameters instead of local model parameters, which reduces both communication costs and privacy risks. Our experiments on benchmark datasets – such as MNIST, CIFAR-10, CIFAR-100, and TinyImageNet – demonstrates a significant enhancement in model convergence and accuracy compared to state-of-the-art solutions, while reducing communication overhead and privacy risks.

## 1 INTRODUCTION

Federated Learning (FL) has gained prominence as an effective approach for collaboratively training machine learning models across decentralized datasets, ensuring data privacy by eliminating the need to share raw data between clients McMahan et al. (2017). Despite its potential, FL performance tends to degrade significantly when data distributions across clients are highly heterogeneous or non-identically distributed (non-IID) Zhao et al. (2018), posing a critical challenge for many real-world applications.

Addressing this issue has sparked substantial research, as recent advancement embraces various strategies, such as variance reduction Acar et al. (2021); Karimireddy et al. (2020), adaptive aggregation Hsu et al. (2019); Reddi et al. (2021); Chen et al. (2023), feature distillation Yang et al. (2023), representation learning Zhang et al. (2020); Tan et al. (2022); Liu et al. (2024) etc., to mitigate the impact of non-IID data on model convergence and performance in FL settings. As these strategies often involve sharing additional information among clients and the server, they lead to increased communication costs and heightened risks of privacy breaches, such as membership inference, features inference, gradient leakage, and model inversion attacks, etc., Nasr et al. (2019); Melis et al. (2019); Wang et al. (2019b). These factors can limit the practical applicability of state-of-the-art approaches, especially in scenarios where communication efficiency and privacy are critical, such as in mobile edge computing Wang et al. (2019a), internet of things (IoT) networks Nguyen et al. (2021), and healthcare applications Xu & Wang (2021). A comparative summary of the communication cost incurred by state-of-the-art approaches is shown in Table 1. Therefore, there is a clear need for a holistic approach that improves performance in heterogeneous settings while addressing communication overhead and enhancing privacy protection in the FL system.

In response to this, we present FLAIR (**F**ederated **L**earning with **A**ugmented and **I**mproved feature **R**epresentations), a novel FL framework designed to systematically guide local training based

| Method | Local Model sharing | Sharing of Additional Information | Communication Cost per Round |
|---|---|---|---|
| FedAvg McMahan et al. (2017) | ✓ | ✗ | $\mathcal{O}(S_t \times M)$ |
| FedAvgM Hsu et al. (2019) | ✓ | ✗ | $\mathcal{O}(S_t \times M)$ |
| FedProx Li et al. (2020) | ✓ | ✗ | $\mathcal{O}(S_t \times M)$ |
| FedFA Zhang et al. (2020) | ✓ | ✗ | $\mathcal{O}(S_t \times M)$ |
| SCAFFOLD Karimireddy et al. (2020) | ✓ | Control variates ($V$) | $\mathcal{O}(S_t \times (M+V))$ |
| FedProto Tan et al. (2022) | ✗ | Global Protos ($P$), Protos ($\tilde{P}$) | $\mathcal{O}(S_t \times (P+\tilde{P}))$ |
| Elastic Chen et al. (2023) | ✓ | Layer-wise sensitivity ($L$) | $\mathcal{O}(S_t \times (M+L))$ |
| FedFed Yang et al. (2023) | ✓ | Global shared features ($F$), Local sensitive features ($\tilde{F}$) | $\mathcal{O}(S_t \times M + K \times (F+\tilde{F}))$ |
| FLUTE Liu et al. (2024) | ✓ | Local classifier weight $C$ | $\mathcal{O}(S_t \times (M+C))$ |
| FLAIR | ✗ | CVAE Parameters $E$ | $\mathcal{O}(E)$ |

Table 1: Comparative summary of federated learning approaches in terms of key attributes and communication cost per round ($K$: total number of clients, $S_t$: number of local models, $M$: size of the model parameters)

on class-oriented features generated from conditional variational autoencoders (CVAE) Sohn et al. (2015). This approach effectively addresses issues of class imbalance and missing classes while also reducing communication costs and enhancing privacy. In particular, FLAIR adopts the following key strategies:

1. **Feature Augmentation:** Leveraging CVAE, we generate synthetic feature samples that enable clients to learn class-specific representations, improving overall feature extraction and representation learning.

2. **Classifier Tuning:** The CVAE-based framework allows clients to generate features for all (including missing) classes, addressing issues of class imbalance and the absence of certain classes in local datasets due to extreme non-IID distributions.

3. **Knowledge Transfer:** While CVAE can generate features for a specific round, they may struggle to adapt to evolving feature representations during local model updates. Reptile meta-training approach Nichol et al. (2018) helps bridge this gap by enabling efficient transfer of knowledge between previous and updated CVAE, ensuring consistency in feature generation despite changes in local models.

The novelty of our approach lies in the synergistic combination of feature generation modeling and representation learning techniques. This holistic strategy not only addresses the symptoms of non-IID data (e.g., model divergence) but also improves communication overhead and privacy of clients.

To summarize, the primary contributions of this paper are as follows:

1. We propose FLAIR, a novel approach that addresses the challenges of learning from extreme non-IID data distributions in federated settings, which reduces communication overhead and privacy risks.

2. We develop a CVAE-based local feature augmentation strategy to generate synthetic features following local class distributions, mitigating class imbalance and missing class issues.

3. We adopt Reptile meta-training approach in FL to mitigate dynamic features drift for stabilizing CVAE model training.

4. We demonstrate FLAIR's superior performance compared to state-of-the-art methods through extensive experiments on benchmark datasets.

The rest of this paper is organized as follows: Section 2 provides a detailed description of our proposed approach. Section 3 presents theoretical analysis of FLAIR, proving its convergence and robustness. Section 4 demonstrates performance evaluation of FLAIR, along with comparative summary w.r.t. state of the art works. Section 5 discusses the state-of-the-artworks. Finally, Section 6 concludes our work.

## 2 FLAIR: PROPOSED FL APPROACH

The primary objective of FLAIR is to maintain a robust and generalized global model in the presence of extreme heterogeneous settings, while maintaining a balance between communication costs and

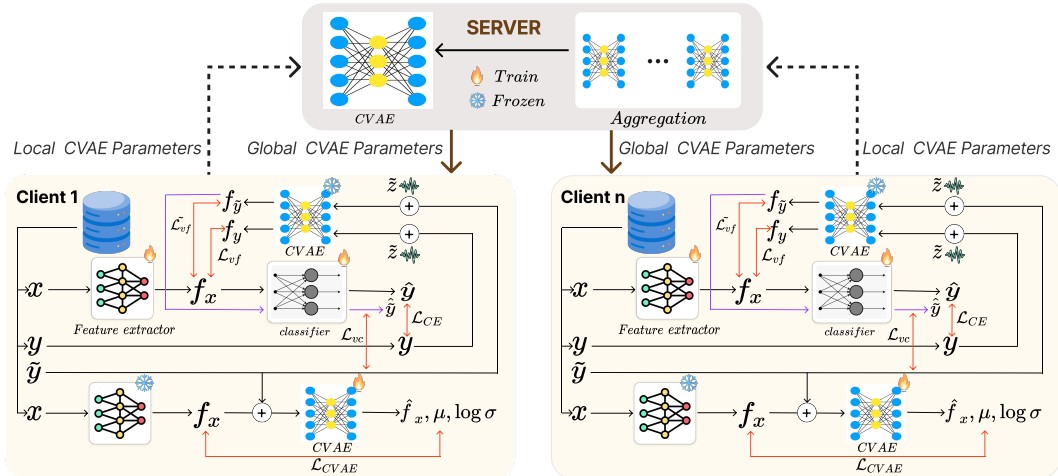

Figure 1: FLAIR architecture

privacy measures. The clients in FLAIR are responsible for locally train the global and CVAE model, whereas the server is responsible for aggregating locally trained CVAE model. Let us describe this process in detail:

## 2.1 TRAINING OBJECTIVES

Consider a federated learning setting with a set of clients $\mathbb{K}$, where each client $k \in \mathbb{K}$ has a local dataset $\mathbb{D}_k = \{(x_k^i, y_k^i)\}_{i=1}^{n_k}$, where $n_k$ is the total number of data samples in client $k$. The data samples $x_k^i \in \mathcal{X}$ and labels $y_k^i \in \mathcal{Y} = \{1, \ldots, C\}$ are drawn from client-specific distributions $p_k(x, y)$, where $C$ is the total number of unique classes among all clients. In the non-IID setting, the distributions $p_k$ can differ significantly across clients. The goal is to learn a generalized and robust model $\mathcal{F}(\Theta, \mathcal{X}) \to \mathcal{Y}$ parameterized by $\Theta$ that performs well on all clients' data distributions:

$$\min_{\Theta} \sum_{k \in \mathbb{K}} \mathbb{E}_{(x,y) \sim p_k}[\mathcal{L}(\mathcal{F}(\Theta, x), y)] \tag{1}$$

where $\mathcal{L} : \hat{\mathcal{Y}} \times \mathcal{Y} \to \mathbb{R}_+$ is the objective function, e.g. cross-entropy loss for classification task.

## 2.2 LOCAL TRAINING

In each round $t$, the server randomly selects a set of clients $\mathbb{S}_t \subseteq \mathbb{K}$ from the pool of available clients. The server then distributes the global CVAE model parameters $(\Phi_t, \Psi_t)$ to all the selected clients. Each client $k \in \mathbb{S}_t$ then sets its local CVAE model parameters as $\phi_{k,t} \leftarrow \Phi_t, \psi_{k,t} \leftarrow \Psi_t$. Subsequently, for each epoch $e$ with learning rate $\eta_l$, the clients update their local model parameters by minimizing the local objective, as follows:

$$\theta_{k,t}^{e+1} = \theta_{k,t}^e - \eta_l \nabla_{\theta_{k,t}^e} \mathcal{L}_C(\theta_{k,t}^e, x, y) \tag{2}$$

Here, $\mathcal{L}_C$ is a combined loss function defined as:

$$\mathcal{L}_C(\theta_{t,k}^e, x, y) = \mathcal{L}_{CE}(\theta_{t,k}^e, x, y) + \lambda_f \mathcal{L}_{vf}(\theta_{t,k}^e, f_x, f_y) \tag{3}$$
$$+ \lambda_{\tilde{f}} \tilde{\mathcal{L}}_{vf}(\theta_{t,k}^e, f_x, f_{\tilde{y}}) + \lambda_c \mathcal{L}_{vc}(\theta_{t,k}^e, f_{\tilde{y}}, \tilde{y})$$

where $\mathcal{L}_{CE}(\theta_{t,k}^e, x, y)$ is the cross-entropy (CE) loss between the predicted and target values, $\mathcal{L}_{vf}(\theta_{t,k}^e, f_x, f_y)$ is the mean-squared error (MSE) loss between the model features $f_x$ and CVAE generated features $f_y$ to enhance intra-class consistency, $\tilde{\mathcal{L}}_{vf}(\theta_{t,k}^e, f_x, f_{\tilde{y}})$ is the negative MSE loss between the model features $f_x$ and CVAE generated features $f_{\tilde{y}}$ to increase inter-class separation, and $\mathcal{L}_{vc}(\theta_{t,k}^e, f_{\tilde{y}}, \tilde{y})$ is the CE loss between the predicted label of features $f_{\tilde{y}}$ generated by the CVAE

and the target label $\tilde{y}$. $\lambda_f$, $\lambda_{\tilde{f}}$, and $\lambda_c$ are hyper-parameters that control the relative importance of the $\mathcal{L}_{vf}$, $\tilde{\mathcal{L}}_{vf}$ and $\mathcal{L}_{vc}$ terms, respectively. Here $\tilde{y} \in \{1, .., C\}|\tilde{y} \neq y$, is a randomly selected class from all possible classes, with the constraint that $\tilde{y} \neq y$. This random class selection helps the CVAE learn to generate diverse and representative features for different classes including missing classes.

After locally updating the client model using global CVAE parameters, the client adapts to the feature shift by fine-tuning the CVAE model following Reptile meta-training approach. Finally, the clients share their updated CVAE model parameters with the server. Ovserve that, in FLAIR, the CVAE model emphasizes feature-level reconstruction while injects noise into latent input features generated by local model. Therefore, the sharing of CVAE model parameters significantly reduces the exposure of sensitive information and effectively mitigates various privacy attacks while maintaining high performance.

## 2.3 CVAE FOR FEATURE GENERATION

To alleviate the impact of non-IID data in federated learning, we propose training a CVAE model on each client's local dataset to model the class-conditional feature distributions $p(\mathbf{z}|f_x, y)$ and $p(f_y|\mathbf{z}, y)$. The CVAE consists of an encoder network $q_\phi(\mathbf{z}|f_x, y)$ and a decoder network $p_\psi(f_y|\mathbf{z}, y)$, parameterized by $\phi$ and $\psi$, respectively. Given an input feature vector $f_x \in \mathbb{R}^d$ and its corresponding label $y \in 1, \ldots, C$, the encoder maps $(f_x, y)$ to a latent code $\mathbf{z} \in \mathbb{R}^l$, where $l$ is the dimensionality of the latent space. The latent code $\mathbf{z}$ is assumed to follow a multivariate Gaussian distribution $\mathcal{N}(\boldsymbol{\mu}, \mathrm{diag}(\boldsymbol{\sigma}^2))$, where the mean $\boldsymbol{\mu} \in \mathbb{R}^l$ and variance $\boldsymbol{\sigma}^2 \in \mathbb{R}^l$ are outputs of the encoder network. The decoder network takes as input a latent code $\mathbf{z}$ sampled from $\mathcal{N}(\boldsymbol{\mu}, \mathrm{diag}(\boldsymbol{\sigma}^2))$ and the label $y$, and reconstructs the input feature vector $\hat{f}_x$. The goal is to maximize the likelihood of the input features given the latent code and labels, i.e. $p_\psi(f_x|\mathbf{z}, y)$. For each client $k$, the CVAE model is trained by maximizing the evidence lower bound (ELBO) of the log-likelihood:

$$\mathcal{L}_{\mathsf{CVAE}} = \mathbb{E}_{(\mathbf{x},y)\sim\mathbb{D}_k}\left[\mathbb{E}_{q_{\phi_{k,t}}(\mathbf{z}|x,y)}\left(\frac{1}{d}\sum_{i=1}^{d}(f_{x_i} - \hat{f}_{x_i})^2\right) - \mathrm{KL}\left(q_{\phi_{k,t}}(\mathbf{z}|x,y)\|p(\mathbf{z}|y)\right)\right] \quad (4)$$

where $\mathbb{D}_k$ denotes the data distribution for client $k$, while $(\phi_{k,t}, \psi_{k,t})$ represent the parameters of the CVAE encoder and decoder for client $k$, respectively, and $p(\mathbf{z}|y)$ is the prior distribution over the latent codes for each class, typically chosen to be a standard Gaussian $\mathcal{N}(\mathbf{0}, \mathbf{I})$. The first term in the ELBO is the reconstruction loss, which encourages the decoder to accurately reconstruct the input features. The second term is the Kullback-Leibler (KL) divergence between the posterior distribution $q_{\phi_{k,t}}(\mathbf{z}|f_x, y)$ and the prior $p(\mathbf{z}|y)$, which acts as a regularizer to prevent over-fitting. In practice, the ELBO is optimized using stochastic gradient descent, with the reconstruction loss approximated by the MSE between the input and reconstructed features:

$$\mathcal{L}_{\mathsf{MSE}}(f_x, \hat{f}_x) = \frac{1}{d}\sum_{i=1}^{d}(f_{x_i} - \hat{f}_{x_i})^2 \quad (5)$$

and the KL divergence computed analytically for Gaussian distributions:

$$\mathcal{L}_{\mathsf{KLD}}(\mu, \log\sigma^2) = \mathrm{KL}(\mathcal{N}(\boldsymbol{\mu}, \mathrm{diag}(\boldsymbol{\sigma}^2))|\mathcal{N}(\mathbf{0}, \mathbf{I}))$$

$$= \frac{1}{2}\sum_{i=1}^{l}(\mu_i^2 + \sigma_i^2 - \log\sigma_i^2 - 1) \quad (6)$$

The overall training objective is a weighted combination of the reconstruction loss, KL divergence, and center loss:

$$\mathcal{L}_{\mathsf{CVAE}}(x, \hat{x}, \mu, \log\sigma^2, z, c) = \mathcal{L}_{\mathsf{MSE}}(f_x, \hat{f}_x) + \lambda \cdot \mathcal{L}_{\mathsf{KLD}}(\mu, \log\sigma^2) \quad (7)$$

where $\lambda$ is the hyper-parameter to control KL divergence. After training the CVAE on its local data, each client can generate synthetic features $f_y$ by first sampling a latent code $\mathbf{z} \sim p(\mathbf{z}|\tilde{y}) = \mathcal{N}(\mathbf{0}, \mathbf{I})$ for a desired class $y$ and a random latent space $\tilde{\mathbf{z}}$, and then decoding it using the trained decoder network: $f_y \sim p_\psi(\hat{f}_x|\tilde{\mathbf{z}}, y)$. These generated samples is used to maintain consistency between

---

**Algorithm 1** Reptile-based CVAE Training

1: Input: Global CVAE parameters $(\Phi_t, \Psi_t)$, Local dataset $\mathbb{D}_k$, Hyper-parameters: $(\alpha)$
2: Output: Client's updated CVAE parameters $(\phi_{k,t+1}, \psi_{k,t+1})$
3: Initialize: Set client's CVAE parameters: $\phi_{k,t} \leftarrow \Phi_t, \psi_{k,t} \leftarrow \Psi_t$
4: **for** each local epoch $e = 1, 2, \ldots, E$ **do**
5:     **for** each batch $(x, y) \in \mathbb{D}_k$ **do**
6:         $f_x \leftarrow \mathcal{F}(\theta_{k,t+1}, x)$
7:         $\hat{f}_x, \mathbf{z}, \hat{y} \leftarrow \mathsf{CVAE}(\phi_{k,t}, f_x, y)$
8:         Compute $\mathcal{L}_{CVAE}$ following Equation 4.
9:         $\phi_{k,t}^{updated} \leftarrow \phi_{k,t}^{old} - \alpha \nabla_{\phi_{k,t}^{old}} \mathcal{L}_{CVAE}$
10:         $\psi_{k,t}^{updated} \leftarrow \psi_{k,t}^{old} - \alpha \nabla_{\psi_{k,t}^{old}} \mathcal{L}_{CVAE}$
11:     **end for**
12: **end for**

---

intra-class and separation among inter-class additionally mitigate class imbalance. At each communication round $t$, client $k$ generates $m_t$ samples per class from its CVAE to obtain an augmented features $\mathbb{F}_{k,t}^{aug} = \mathbb{F}_{k,t} \cup \{(\hat{f}_{x_j}, y_j) | j \in 1 \cdot m_t.C\}$. The use of class-conditional priors $p(\mathbf{z}|y)$ allows the CVAE to learn a separate latent space for each class, enabling it to capture class-specific features and variations. This is particularly beneficial in the federated learning setting, where the data is often non-IID across clients. By generating diverse synthetic samples that follow the local class distributions, the CVAE can help regularize the local models and improve their generalization to unseen data.

## 2.4 REPTILE-BASED CVAE MODEL TRAINING

To address the challenge of dynamic feature shifts in federated learning, we utilize Reptile meta-learning based CVAE model training. This technique is employed after each client's local update to adapt the CVAE model to the local data distribution while preserving the global knowledge.

The Reptile algorithm is a first-order meta-learning approach that aims to find a good initialization of model parameters that can quickly adapt to new tasks with a few gradient steps. In the context of federated learning, we treat each client's local data as a separate task and use Reptile to learn a meta-initialization of the CVAE parameters that can rapidly adapt to the local data distributions. For a client $k$ and its local dataset $\mathbb{D}_k$, the Reptile-based CVAE training proceeds as follows:

In each communication round $t$, the server sends the global CVAE parameters $\Phi_t, \Psi_t$ to the selected clients. Each client $k$ initializes its local CVAE parameters $\phi_{k,t}, \psi_{k,t}$ with $\Phi_t, \Psi_t$ and performs $E$ epochs of training on its local dataset $\mathbb{D}_k$. During CVAE model training, the client's locally trained model $\mathcal{F}(\theta_{k,t+1})$ is used to extract local features $f_x$ from the input data $x$, and the CVAE model takes the features $f_x$ and labels $y$ as input to reconstruct the features $\hat{f}_x$, generate latent variables $\mathbf{z}$, and predict the labels $\hat{\mathbf{y}}$. The CVAE parameters $\phi_{k,t}, \psi_{k,t}$ are updated by minmizing $\mathcal{L}_{CVAE}$ loss. The overall training process is depicted in Algorithm 1.

The Reptile-based CVAE training allows the model to adapt to the local data distributions of each client while maintaining the global knowledge learned across all clients. This approach helps mitigate the impact of dynamic feature shifts and enables more effective federated learning in non-IID settings.

## 2.5 OVERALL TRAINING PROCESS

Algorithm 2 outlines the overall FLAIR training process across multiple clients. During the initialization phase (lines 6-11), each client trains its local CVAE model and shares its parameters with the server. Subsequently, the server aggregates the client parameters into its CVAE model (lines 12, 13). For each communication round, a set of participating clients is randomly sampled (line 16) to engage in local training. These selected clients train in parallel, updating their local models using their respective datasets (lines 19-24). After local model training, each client utilizes the features generated by its updated model to refine its local CVAE model using a Reptile-based approach (as detailed

---

**Algorithm 2** FLAIR: Federated Learning with Augmented and Improved Representations

---

1: **Input:** Number of clients $\mathbb{K}$, number of communication rounds $T$, local datasets $\{\mathbb{D}_k | k \in \mathbb{K}\}$, local model learning rate $\eta_l$, CVAE learning rate $\eta_{CVAE}$, Clients initial CVAE model parameters $(\phi_{k,0}, \psi_{k,0} | k \in \mathbb{K})$

2: **Output:** Generalized and robust local model parameters $(\Theta_k | k \in \mathbb{K})$

3: **Initialization:**

4: Compute initial global CVAE parameters $(\Phi_1, \Psi_1)$:

5: **for** each client $k \in \mathbb{K}$ **in parallel do**

6:     **for** epoch $e = 1, \ldots, E_{CVAE}$ **do**

7:         Compute $\mathcal{L}_{CVAE}$ following Equation 4.

8:         $\phi_{k,0}^{e+1} \leftarrow \phi_{k,0}^e - \eta_{CVAE} \nabla_{\phi_{k,0}^e} \mathcal{L}_{CVAE}$

9:         $\psi_{k,0}^{e+1} \leftarrow \psi_{k,0}^e - \eta_{CVAE} \nabla_{\psi_{k,0}^e} \mathcal{L}_{CVAE}$

10:     **end for**

11: **end for**

12: $\Phi_1 \leftarrow \frac{1}{|\mathbb{K}|} \sum_{k \in \mathbb{K}} \phi_{k,1}$

13: $\Psi_1 \leftarrow \frac{1}{|\mathbb{K}|} \sum_{k \in \mathbb{K}} \psi_{k,1}$

14: **Federated Training:**

15: **for** each communication round $t = 1, \ldots, T$ **do**

16:     **Select** a set of clients $\mathbb{S}_t$ for local training.

17:     **Local Training:**

18:     **for** each client $k \in \mathbb{S}_t$ **in parallel do**

19:         **for** epocs $e = 1, \ldots, E$ **do**

20:             **Sample** a batch of local dataset $\{x, y\} \in \mathbb{D}_k$

21:             **Compute** local features $f_x$

22:             **Generate** synthetic features $f_y, f_{\tilde{y}}$

23:             **Update** local model parameters following Equation 2.

24:         **end for**

25:         **Update** local CVAE parameters $(\phi_{k,t}, \psi_{k,t})$ following Algorithm 1.

26:         **Send** updated $(\phi_{k,t+1}, \psi_{k,t+1})$ to the server.

27:     **end for**

28:     **Aggregation:**

29:     Aggregate **CVAE** parameters:

30:     $\Phi_{t+1} \leftarrow \frac{1}{|\mathbb{S}_t|} \sum_{k \in \mathbb{S}_t} \phi_{k,t+1}$

31:     $\Psi_{t+1} \leftarrow \frac{1}{|\mathbb{S}_t|} \sum_{k \in \mathbb{S}_t} \psi_{k,t+1}$

32: **end for**

---

in Algorithm 1). The updated local CVAE model parameters are then shared with the server for aggregation. Finally, the server aggregates all the shared CVAE parameters of clients (lines 34 and 35). This process is iteratively repeated for each communication round, until it reaches convergence criteria.

## 3 THEORETICAL ANALYSIS OF FLAIR

This section presents the following theorems, which collectively demonstrate the theoretical foundations of FLAIR. **Due to space constraints, the detailed proofs are reported in Appendix A.**

1. **Theorem 1 (Convergence of FLAIR)** shows that both the client models and the global CVAE converge to their respective optimal parameters.

2. **Theorem 2 (Generalization Bound for FLAIR)** provides a bound on the generalization error, taking into account the effect of CVAE-based augmentation.

3. **Theorem 3 (Feature Diversity)** establishes that the features generated by the CVAE are close to the true distribution of features for each class across all clients.

4. **Theorem 4 (Client Model Robustness)** demonstrates that the global CVAE helps in making client models more robust and consistent, even when faced with test distributions that may differ from their training distributions.

These guarantees the effectiveness of FLAIR in addressing the challenges of federated learning in non-IID settings, particularly in terms of improving generalization and robustness across heterogeneous client data distributions.

# 4 EXPERIMENT EVALUATION

## 4.1 IMPLEMENTATION

We implement **FLAIR** and the baseline methods using Python 3.9, leveraging the PyTorch library[1]. The codebase consists of 7,718 lines of code (LoC). For local training, we utilized two distinct neural network architectures: LeNet-5 and ResNet-18. The details of their architectures are as follows:

- **LeNet-5**: A 7-layer convolutional neural network (CNN) featuring 5x5 convolutional layers, tanh activations, and average pooling.
- **ResNet-18**: A CNN beginning with a 7x7 convolutional layer, followed by 4 residual blocks, batch normalization, ReLU activation, and global average pooling.

In addition to local classification models, we have a CVAE model meant to learn enhanced features representation. The CVAE model architecture consists of two components encoder and decoder described as follows:

- **Encoder**: Two fully connected layers with batch normalization and ReLU activations. Input is the concatenation of input features and one-hot encoded class labels.
- **Decoder**: Two fully connected layers, batch normalization, ReLU activations, and an output layer for reconstruction.

We employ the PyTorch SGD optimizer for updating model parameters during training.

## 4.2 EXPERIMENT SETUP

In this study, we conduct a comprehensive evaluation of FLAIR, by comparing its performance against state-of-the-art approaches. To this aim, we consider the following 6 baseline methods: FedAvg, SCAFFOLD, FedFA, FedProto, Elastic, and FLUTE, which represent a diverse range of strategies for federated learning. Our experiments are performed on three widely-used datasets: MNIST, CIFAR-10, CIFAR-100, and TinyImageNet, each presenting unique challenges and characteristics. To investigate the effectiveness of FLAIR and the baseline methods under different model architectures, we employ a variety of neural networks for local training. Specifically, when training on the MNIST dataset, we utilize LeNet-5 model, which is well-suited for the task of handwritten digit recognition. For more complex CIFAR-10, CIFAR-100, and TinyImageNet datasets, we employ ResNet-18 model, which already depicted a superior performance on image classification tasks. To ensure fairness and reproducibility of our comparisons, we maintain fixed random seeds and consistent settings across all experiments. This allows us to isolate the impact of federated learning algorithms on model performance, minimizing the influence of random variations. For all the experiments we use local model learning rate $\eta_l = 0.01$, batch size 16, number of local epochs 5, and for FLAIR's CVAE model training we use Adam optimizer with learning rate $\eta_{CVAE} = 0.001$. We conduct 150 communication rounds FL training for MNIST, 250 rounds for CIFAR-10, and 200 rounds for CIFAR-100 and TinyImageNet dataset.

## 4.3 DATASET DISTRIBUTION

To evaluate the performance under various heterogeneous settings, we establish the following three distinct configurations of dataset distribution:

- *Label Skew*: In this setting, the label distribution varies across clients, simulating a scenario where each client has a different proportion of samples from each class. To create a label-skewed dataset, we use the Dirichlet distribution on the label ratios to ensure uneven label

---

[1]The code for our proposal can be found at: https://anonymous.4open.science/r/FLAIR-C512

| Dataset | Method | Beta | Test Average Acc | | | # of Comm |
|---------|--------|------|--------------------|---|---|-----------|
| | | $\beta$ | $n_s = 0$ | $n_s = 0.1$ | $n_s = 0.2$ | Rounds |
| MNIST | FedAvg | 0.5 | 96.82 | 96.80 | 96.43 | 150 |
| | | 0.05 | 94.39 | 94.10 | 93.75 | 150 |
| | SCAFFOLD | 0.5 | 98.53 | 98.37 | 98.24 | 150 |
| | | 0.05 | 98.45 | 98.39 | 98.37 | 150 |
| | FedFA | 0.5 | 96.80 | 96.77 | 96.43 | 150 |
| | | 0.05 | 94.32 | 94 | 93.7 | 150 |
| | FedProto | 0.5 | 98.71 | 98.76 | 98.71 | 150 |
| | | 0.05 | 98.26 | 98.19 | 98.12 | 150 |
| | Elastic | 0.5 | 97.31 | 97.26 | 96.89 | 150 |
| | | 0.05 | 94.92 | 94.41 | 94.06 | 150 |
| | FLAIR | 0.5 | **98.83** | **98.56** | **98.49** | 150 |
| | | 0.05 | **98.59** | **98.55** | **98.52** | 150 |
| CIFAR10 | FedAvg | 0.5 | 61.97 | 53.02 | 50.55 | 250 |
| | | 0.05 | 40.74 | 27.10 | 23.90 | 250 |
| | SCAFFOLD | 0.5 | 78.19 | 70.28 | 63.45 | 250 |
| | | 0.05 | 41.26 | 35.67 | 26.32 | 250 |
| | FedFA | 0.5 | 60.99 | 53.18 | 45.10 | 250 |
| | | 0.05 | 39.04 | 25.12 | 19.02 | 250 |
| | FedProto | 0.5 | 78.77 | 71.55 | 65.30 | 250 |
| | | 0.05 | 49.34 | 35.54 | 26.29 | 250 |
| | Elastic | 0.5 | 62.45 | 53.36 | 45.75 | 250 |
| | | 0.05 | 42.25 | 29.36 | 25.56 | 250 |
| | FLAIR | 0.5 | **79.36** | **73.06** | **65.52** | 250 |
| | | 0.05 | **53.68** | **38.29** | **30.26** | 250 |
| CIFAR100 | FedAvg | 0.5 | 21.68 | 17.24 | 15.56 | 200 |
| | | 0.05 | 20.52 | 15.43 | 11.87 | 200 |
| | SCAFFOLD | 0.5 | 45.66 | 36.75 | 27.28 | 200 |
| | | 0.05 | 34.16 | 25.72 | 18.16 | 200 |
| | FedFA | 0.5 | 21.52 | 19.84 | 17.96 | 200 |
| | | 0.05 | 20.69 | 15.34 | 11.73 | 200 |
| | FedProto | 0.5 | 44.94 | 35.08 | 25.21 | 200 |
| | | 0.05 | 34.48 | 24.44 | 18.32 | 200 |
| | Elastic | 0.5 | 22.13 | 18.76 | 16.32 | 200 |
| | | 0.05 | 23.20 | 15.87 | 12.24 | 200 |
| | FLAIR | 0.5 | **47.89** | **37.69** | **27.53** | 200 |
| | | 0.05 | **38.79** | **26.86** | **22.34** | 200 |

Table 2: Performance comparison on four benchmark datasets with varying beta and noise levels. The best results for each dataset and configuration are in highlighted in bold.

distributions among clients. The parameter $\beta$ of the Dirichilet distribution decides the extent of the skew. For our experiments we set $\beta$ to 0.5 and 0.05.

- *Quantity Skew*: In quantity skew, the size of the local dataset varies across parties, although data distribution may still be consistent among the parties. Like distribution-based label skew setting, we use Dirichlet distribution to allocate different amounts of data samples into each party.

- *Feature Skew*: In feature distribution skew, the feature distributions $P(x_i)$ vary across parties although the knowledge $P(y_i|x_i)$ is same. Here we use noise based feature skew with other non IID configurations. In noise based skew, we distort the data slightly by adding different levels of Gaussian noise into it. The intensity of the noise label can be controlled by changing the coefficient associated with the Gaussian noise. We set the noise coefficient $n_s$ to 0, 0.1 or 0.2 for our experiments.

## 4.4 PERFORMANCE AND PRIVACY ANALYSIS

In this section we evaluate the performance of our proposed approach, FLAIR, against state-of-the-art federated learning algorithms such as FedAvg, SCAFFOLD, FedFA, FedProto, and Elastic. The experiments are conducted using datasets with distribution-based label imbalance, generated through Beta values of 0.5 (mild heterogeneity) and 0.05 (extreme heterogeneity), with added Gaussian noise levels of 0, 0.1, and 0.2 to simulate feature skewness, as depicted in Table 2.

The results demonstrate that FLAIR consistently outperforms the state of the art methods across various datasets, with the performance gap widening as dataset complexity and heterogeneity increase. For example, while the performance boost of FLAIR on the MNIST dataset is marginal, it becomes more pronounced on CIFAR-10 and even larger as heterogeneity (controlled by the Beta value) rises. Furthermore, FLAIR is robust in the presence of increasing noise, making it well-suited

| Method | MIA ($\downarrow$) | Gradient Leak ($\downarrow$) | Model Inversion ($\downarrow$) | Info Theoretic ($\downarrow$) |
|--------|------|--------------|----------------|----------------|
| FedProto | ✗ | ✗ | 1.6876 | ✗ |
| Scaffold | 0.4982 | 0.3156 | 1.5937 | 1.5328 |
| FedAvg | 0.4948 | 0.2817 | 1.9949 | 0.1192 |
| Elastic | 0.5088 | 0.3217 | 2.0949 | 0.1282 |
| FedFA | 0.4973 | 0.2719 | 1.9393 | 0.1406 |
| FLAIR | ✗ | ✗ | 0.3103 | ✗ |

Table 3: Privacy Measure Metrics for Various Federated Learning Approaches, here ✗ denotes that the privacy measure is not available

for scenarios with extreme data distributions, both in terms of noise and heterogeneity. For instance, on CIFAR-10 dataset, when $\beta = 0.05$ and noise level 0.2 we get average test accuracy of FLAIR as **30.26**, while the accuracy of the second best model, SCAFFOLD is **26.32**. This type of noticeable jumps is seen in the results in almost all cases which states the effectiveness of our proposed FLAIR approach.

The superior performance of FLAIR is largely attributed to its use of CVAE-based class feature representations, which, in a federated learning setting, can effectively approximate the feature distribution for each class. In contrast, methods like FedAvg, SCAFFOLD, and FedFA exhibit significant performance degradation as the data becomes more heterogeneous. While FedProto, leveraging class prototype-based representations, manages to mitigate some distribution challenges, FLAIR outperforms it in nearly every test case due to the strength of its CVAE-based approach. These results affirm the suitability of FLAIR for complex and imbalanced federated learning environments. Additional results are available in the Appendix B of the paper.

Table 3 shows a strong evidence in support of FLAIR's privacy preserving capabilities across multiple metrics. It is immune to Membership Inference Attacks (MIA) and Gradient Leak attacks, as indicated by the ✗ symbols. In the Model Inversion metric, FLAIR achieves the lowest score of 1.03, significantly outperforming other methods. The absence of an Info Theoretic score for FLAIR (indicated by ✗) suggests that it does not leak information through this channel. The primary reason behind this success is due to sharing of only locally trained **CVAE** model parameters, rather than local model parameters, as described in Section 2. In particular, through the injection of noise into the latent features, FLAIR substantially reduces the risk of inversion attacks, making exact recovery of original data infeasible. The generalization of feature reconstructions further mitigates membership inference attacks by minimizing overfitting to specific data points. Moreover, the absence of gradient and raw data sharing across the network significantly diminishes the attack surface for gradient-based exploits. This synergistic combination of noise injection, feature-level focus, and non-sharing of gradients establishes a robust privacy-preserving mechanism, effectively balancing collaborative learning with stringent privacy requirements in federated learning applications.

To summarize, FLAIR demonstrates superior performance and enhanced privacy protection in complex, imbalanced federated learning environments. It consistently outperforms existing methods across various datasets and heterogeneity levels while providing robust privacy guarantees. These results highlight the effectiveness of FLAIR's CVAE-based approach in addressing both performance and privacy challenges in federated learning.

## 5 RELATED WORK

Federated learning was first introduced in FedAvg McMahan et al. (2017) algorithm. This method allows clients to collaboratively train models without directly sharing their data, thus preserving privacy. However, despite the inherent privacy benefits, it presents several key challenges, particularly high communication costs and instability in model training due to data heterogeneity across clients. FedAvgM Hsu et al. (2019) integrates momentum into the global model updates, speeding up convergence in non-IID settings. While this reduces the number of communication rounds required, it introduces the need for careful hyperparameter tuning to maintain stability in diverse data environments. FedProx Li et al. (2020) introduced a proximal term in the local objective function to reduce client drift, which indirectly reduces communication by requiring fewer updates to converge. However, FedProx still struggles in extreme non-IID distribution settings. SCAFFOLD Karimireddy

et al. (2020) attempts to directly address client drift using control variates to stabilize local updates, achieving better convergence with fewer rounds of communication, albeit with an increase in per-round communication overhead due to additional information exchanged between clients and server. Eastic Chen et al. (2023) improves convergence through aggregation by interpolating client models according to its parameter sensitivity. However this requires sharing of additional client's parameters sensitivity with the server and struggle in extreme heterogeneous settings. FedFA Zhou et al. (2024) is mainly used to address the issue of feature skewness by utilizing feature anchors but struggles in others heterogeneous settings. FedProto Tan et al. (2022) addresses data heterogeneity in federated learning by aligning global feature distributions across clients. While this approach effectively improves model performance, it shares class-specific features, potentially exposing sensitive class information and making the system vulnerable to privacy attacks, such as feature inversion Wang et al. (2019b). FedBN Li et al. (2021), on the other hand, applies client-specific batch normalization layers to handle feature skew, but fails to generalize in settings with severe data imbalance, leading to increased communication rounds as the global model struggles to converge. Recent works such as FedFed Yang et al. (2023) and FLUTE Liu et al. (2024) explore more advanced techniques to address data heterogeneity. FedFed enhances model accuracy by performing feature augmentation across clients but introduces significant communication and computational overhead due to the need for sharing additional feature information. Similarly, FLUTE employs feature learning and classifier calibration to address heterogeneity, but it suffers from the need for extensive hyperparameter tuning and full client participation, which leads to increased communication demands. In summary, while most of the existing approaches primarily focus in addressing data heterogeneity, they often introduce increased communication costs and heightened risks of privacy breaches.

## 6 CONCLUSION

This paper presents a novel FL approach, aiming to achieve a balance between performance, communication cost, and privacy. This is achieved by utilizing CVAE based feature augmentation approach which helps in developing generalized and robust local models, making them effective in addressing extreme heterogeneity. In particular, the augmented features helps the local models to promote consistency in intra-class latent representations while simultaneously amplifying inter-class distinctions. Further to adapt dynamic feature shifts in CVAE model we utilize Reptile meta-training approach. Unlike existing approaches, the sharing of only CVAE model parameters, rather than local model parameters, reduces privacy risks and communication overhead. Experimental results demonstrates a significant performance improvement by 2.61% on an average in terms of accuracy with respect to the second best performer in the literature. This highlights FLAIR's ability to enhance the performance of underrepresented classes in clients, leading to a more balanced and equitable learning outcome. Furthermore, **FLAIR** exhibits faster convergence rates with reduced communication cost compared to existing methods, maintaining its performance advantages even in extreme heterogeneous settings.

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

APPENDIX

# A THEORETICAL ANALYSIS OF FLAIR

## A.1 PRELIMINARIES AND NOTATION

Let $\mathbb{K}$ be the set of all clients, with $|\mathbb{K}| = K$. For each client $k \in \mathbb{K}$, let $\mathbb{D}_k$ be its local dataset, with $n_k = |\mathbb{D}_k|$ and $n = \sum_{k=1}^{K} n_k$. Let $\theta_{k,t} \in \mathbb{R}^d$ denote the local model parameters of client $k$ at round $t$. Let $\phi_t \in \mathbb{R}^p$ and $\psi_t \in \mathbb{R}^q$ denote the global CVAE encoder and decoder parameters at round $t$, respectively.

We define the following:

- $\mathcal{L}(\theta; x, y)$: loss function for client models
- $\mathcal{L}_{CVAE}(\phi, \psi; x, y)$: loss function for CVAE
- $F_k(\theta_k) = \mathbb{E}_{(x,y) \sim \mathbb{D}_k}[\mathcal{L}(\theta_k; x, y)]$: expected loss for client $k$
- $G(\phi, \psi) = \frac{1}{K} \sum_{k=1}^{K} \mathbb{E}_{(x,y) \sim \mathbb{D}_k}[\mathcal{L}_{CVAE}(\phi, \psi; x, y)]$: expected CVAE loss

## A.2 ASSUMPTIONS

We make the following assumptions:

**Assumption 1 (Smoothness)** $F_k$ *is L-smooth for all k:* $\forall \theta_k, \theta'_k \in \mathbb{R}^d$,

$$\|\nabla F_k(\theta_k) - \nabla F_k(\theta'_k)\| \leq L\|\theta_k - \theta'_k\| \tag{8}$$

**Assumption 2 (Strong Convexity)** $F_k$ *is $\mu$-strongly convex for all k:* $\forall \theta_k, \theta'_k \in \mathbb{R}^d$,

$$F_k(\theta'_k) \geq F_k(\theta_k) + \langle \nabla F_k(\theta_k), \theta'_k - \theta_k \rangle + \frac{\mu}{2}\|\theta'_k - \theta_k\|^2 \tag{9}$$

**Assumption 3 (Bounded Variance)** *The variance of stochastic gradients is bounded for both client models and CVAE:*

$$\mathbb{E}_{(x,y) \sim \mathbb{D}_k}[\|\nabla \mathcal{L}(\theta_k; x, y) - \nabla F_k(\theta_k)\|^2] \leq \sigma^2, \quad \forall k \in \mathbb{K}, \forall \theta_k \in \mathbb{R}^d \tag{10}$$

$$\mathbb{E}_{(x,y) \sim \mathbb{D}_k}[\|\nabla \mathcal{L}_{CVAE}(\phi_k, \psi_k; x, y) - \nabla G(\phi, \psi)\|^2] \leq \sigma_{CVAE}^2, \quad \forall k \in \mathbb{K}, \forall \phi_k, \psi_k \tag{11}$$

**Assumption 4 (CVAE Lipschitz Continuity)** *The CVAE model is Lipschitz continuous with respect to its parameters:*

$$\|\text{CVAE}(x; \phi, \psi) - \text{CVAE}(x; \phi', \psi')\| \leq L_{CVAE}(\|\phi - \phi'\| + \|\psi - \psi'\|) \tag{12}$$

## A.3 CONVERGENCE ANALYSIS

**Lemma 1 (One-step Progress for Client Models)** *For any client $k$ and round $t \geq 0$, its one-step convergence bound follows:*

$$\mathbb{E}[\|\theta_{k,t+1} - \theta_k^*\|^2] \leq (1 - \mu\eta_l)\mathbb{E}[\|\theta_{k,t} - \theta_k^*\|^2] - 2\eta_t\mathbb{E}[F_k(\theta_{k,t}) - F_k(\theta_k^*)]$$
$$+ \eta_l^2\left(\sigma^2 + 6L\gamma E\eta_l + 8(E-1)^2\gamma^2\right) \tag{13}$$

**Proof 1** *Detailed proof is reported in the Appendix .., due to space constraints. Let $g_{k,t}$ be the average stochastic gradient computed by client $k$ at round $t$ over $E$ local epochs. The update rule for client $k$'s model parameters is $\theta_{k,t+1} = \theta_{k,t} - \eta_l g_{k,t}$. We begin by expanding $\|\theta_{k,t+1} - \theta_k^*\|^2$:*

$$\|\theta_{k,t+1} - \theta_k^*\|^2 = \|\theta_{k,t} - \eta_l g_{k,t} - \theta_k^*\|^2$$
$$= \|\theta_{k,t} - \theta_k^*\|^2 - 2\eta_l\langle g_{k,t}, \theta_{k,t} - \theta_k^*\rangle + \eta_l^2\|g_{k,t}\|^2$$

*Taking the expectation of both sides:*

$$\mathbb{E}[\|\theta_{k,t+1} - \theta_k^*\|^2] = \mathbb{E}[\|\theta_{k,t} - \theta_k^*\|^2] - 2\eta_l \mathbb{E}[\langle g_{k,t}, \theta_{k,t} - \theta_k^*\rangle] + \eta_l^2 \mathbb{E}[\|g_{k,t}\|^2] \tag{14}$$

*Using the strong convexity and smoothness of $F_k$, we can bound the inner product term:*

$$\langle g_{k,t}, \theta_{k,t} - \theta_k^*\rangle \geq F_k(\theta_{k,t}) - F_k(\theta_k^*) + \frac{\mu}{2}\|\theta_{k,t} - \theta_k^*\|^2 - \langle g_{k,t} - \nabla F_k(\theta_{k,t}), \theta_{k,t} - \theta_k^*\rangle$$

*For the gradient norm, we use the smoothness property and the definition of $\gamma$:*

$$\mathbb{E}[\|g_{k,t}\|^2] \leq 4L(F_k(\theta_{k,t}) - F_k(\theta_k^*)) + 2\sigma^2 + 2\gamma^2 \tag{15}$$

*Combining these bounds and applying Cauchy-Schwarz inequality to the error term, we get:*

$$\mathbb{E}[\|\theta_{k,t+1} - \theta_k^*\|^2] \leq (1 - 2\mu\eta_l)\mathbb{E}[\|\theta_{k,t} - \theta_k^*\|^2] - 2\eta_l(1 - \eta_l L)\mathbb{E}[F_k(\theta_{k,t}) - F_k(\theta_k^*)]$$
$$+ \eta_l^2(2\sigma^2 + 2\gamma^2) + \eta_l\sigma^2$$

*To account for $E$ local epochs, we model this as $E$ consecutive updates with learning rate $\eta_l$. Applying the above inequality $E$ times and using the convexity of $F_k$, we get the following:*

$$\mathbb{E}[\|\theta_{k,t+E} - \theta_k^*\|^2] \leq (1 - 2\mu\eta_l)^E \mathbb{E}[\|\theta_{k,t} - \theta_k^*\|^2] - 2E\eta_l(1 - \eta_l L)\mathbb{E}[F_k(\theta_{k,t}) - F_k(\theta_k^*)]$$
$$+ E\eta_l^2(2\sigma^2 + 2\gamma^2) + E\eta_l\sigma^2 + 4L\eta_l^2 E(E-1)\gamma^2$$

*Finally, using the fact that $\eta_l \leq 1/(4L)$ (which follows from our choice of $\eta_l$) and combining like terms, we obtain the stated bound:*

$$\mathbb{E}[\|\theta_{k,t+1} - \theta_k^*\|^2] \leq (1 - \mu\eta_l)\mathbb{E}[\|\theta_{k,t} - \theta_k^*\|^2] - 2\eta_l\mathbb{E}[F_k(\theta_{k,t}) - F_k(\theta_k^*)]$$
$$+ \eta_l^2(\sigma^2 + 6L\gamma E\eta_l + 8(E-1)^2\gamma^2) \tag{16}$$

**Lemma 2 (One-step Progress for *CVAE*)** *For any round $t \geq 0$, the convergence bound for *CVAE* follows:*

$$\mathbb{E}[\|(\phi_{t+1}, \psi_{t+1}) - (\phi^*, \psi^*)\|^2] \leq (1 - L_{\text{CVAE}}\eta_{\text{CVAE}})\mathbb{E}[\|(\phi_t, \psi_t) - (\phi^*, \psi^*)\|^2]$$
$$- 2\eta_{\text{CVAE}}\mathbb{E}[G(\phi_t, \psi_t) - G(\phi^*, \psi^*)] + \eta_{\text{CVAE}}^2\sigma_{\text{CVAE}}^2 \tag{17}$$

**Proof 2** *Detailed proof is reported in the Appendix .., due to space constraints. Let $g_t = (\nabla_\phi G(\phi_t, \psi_t), \nabla_\psi G(\phi_t, \psi_t))$ be the gradient of the *CVAE* loss function at round $t$. The update rule for the *CVAE* parameters is:*

$$(\phi_{t+1}, \psi_{t+1}) = (\phi_t, \psi_t) - \eta_{\text{CVAE}} g_t \tag{18}$$

*We begin by expanding $\|(\phi_{t+1}, \psi_{t+1}) - (\phi^*, \psi^*)\|^2$:*

$$\|(\phi_{t+1}, \psi_{t+1}) - (\phi^*, \psi^*)\|^2 = \|(\phi_t, \psi_t) - \eta_{\text{CVAE}} g_t - (\phi^*, \psi^*)\|^2$$
$$= \|(\phi_t, \psi_t) - (\phi^*, \psi^*)\|^2 - 2\eta_{\text{CVAE}}\langle g_t, (\phi_t, \psi_t)$$
$$- (\phi^*, \psi^*)\rangle + \eta_{\text{CVAE}}^2\|g_t\|^2$$

*Taking the expectation of both sides:*

$$
\begin{aligned}
\mathbb{E}[\|(\phi_{t+1}, \psi_{t+1}) - (\phi^*, \psi^*)\|^2] =& \mathbb{E}[\|(\phi_t, \psi_t) - (\phi^*, \psi^*)\|^2] \\
& - 2\eta_{\text{CVAE}}\mathbb{E}[\langle g_t, (\phi_t, \psi_t) - (\phi^*, \psi^*)\rangle] + \eta_{\text{CVAE}}^2\mathbb{E}[\|g_t\|^2]
\end{aligned}
\tag{19}
$$

*By the $L_{\text{CVAE}}$-smoothness of $G$, we have:*

$$
G(\phi^*, \psi^*) \geq G(\phi_t, \psi_t) + \langle g_t, (\phi^*, \psi^*) - (\phi_t, \psi_t)\rangle + \frac{1}{2L_{\text{CVAE}}}\|g_t\|^2
\tag{20}
$$

*Rearranging this inequality:*

$$
\langle g_t, (\phi_t, \psi_t) - (\phi^*, \psi^*)\rangle \geq G(\phi_t, \psi_t) - G(\phi^*, \psi^*) + \frac{1}{2L_{\text{CVAE}}}\|g_t\|^2
\tag{21}
$$

*Substituting Equation 21 into Equation 19, we have:*

$$
\begin{aligned}
\mathbb{E}[\|(\phi_{t+1}, \psi_{t+1}) - (\phi^*, \psi^*)\|^2] \leq& \mathbb{E}[\|(\phi_t, \psi_t) - (\phi^*, \psi^*)\|^2] \\
& - 2\eta_{\text{CVAE}}\mathbb{E}[G(\phi_t, \psi_t) - G(\phi^*, \psi^*)] \\
& - \eta_{\text{CVAE}}\mathbb{E}[\|g_t\|^2]/L_{\text{CVAE}} + \eta_{\text{CVAE}}^2\mathbb{E}[\|g_t\|^2]
\end{aligned}
$$

*Now, using the assumption of bounded variance of stochastic gradients, $\mathbb{E}[\|g_t\|^2] \leq \sigma_{\text{CVAE}}^2$, we get:*

$$
\begin{aligned}
\mathbb{E}[\|(\phi_{t+1}, \psi_{t+1}) - (\phi^*, \psi^*)\|^2] \leq& \mathbb{E}[\|(\phi_t, \psi_t) - (\phi^*, \psi^*)\|^2] \\
& - 2\eta_{\text{CVAE}}\mathbb{E}[G(\phi_t, \psi_t) - G(\phi^*, \psi^*)] \\
& - \eta_{\text{CVAE}}\sigma_{\text{CVAE}}^2/L_{\text{CVAE}} + \eta_{\text{CVAE}}^2\sigma_{\text{CVAE}}^2
\end{aligned}
$$

*Finally, rearranging terms:*

$$
\begin{aligned}
\mathbb{E}[\|(\phi_{t+1}, \psi_{t+1}) - (\phi^*, \psi^*)\|^2] \leq& (1 - \eta_{\text{CVAE}}/L_{\text{CVAE}})\mathbb{E}[\|(\phi_t, \psi_t) - (\phi^*, \psi^*)\|^2] \\
& - 2\eta_{\text{CVAE}}\mathbb{E}[G(\phi_t, \psi_t) - G(\phi^*, \psi^*)] + \eta_{\text{CVAE}}^2\sigma_{\text{CVAE}}^2
\end{aligned}
$$

**Theorem 1 (Convergence of FLAIR)** *Let Assumptions 1-4 hold. Let the learning rates be set as $\eta_l = \frac{2}{\mu(t+\gamma)}$ for client models and $\eta_{\text{CVAE}} = \frac{2}{L_{\text{CVAE}}(t+\gamma)}$ for the **CVAE**, where $\gamma = \max\{8L/\mu, E\}$ and $E$ is the number of local epochs. Then, for $T \geq 1$, the output of **FLAIR** satisfies, following convergence bound:*

$$
\frac{1}{K}\sum_{k=1}^{K}\mathbb{E}[F_k(\bar{\theta}_{k,T})] - F_k(\theta_k^*) \leq \frac{4L\gamma}{\mu T}\left(1 + \log\left(\frac{T}{\gamma} + 1\right)\right)
\tag{22}
$$

$$
\mathbb{E}[G(\bar{\phi}_T, \bar{\psi}_T)] - G(\phi^*, \psi^*) \leq \frac{4L_{CVAE}\gamma}{T}\left(1 + \log\left(\frac{T}{\gamma} + 1\right)\right)
\tag{23}
$$

*where $\bar{\theta}_{k,T} = \frac{1}{T}\sum_{t=1}^{T}\theta_{k,t}$, $\bar{\phi}_T = \frac{1}{T}\sum_{t=1}^{T}\phi_t$, $\bar{\psi}_T = \frac{1}{T}\sum_{t=1}^{T}\psi_t$, and $\theta_k^*, \phi^*, \psi^*$ are the respective optimal parameters.*

**Proof 3 (Theorem 1)** *Detailed proof is reported in the Appendix .., due to space constraints. We prove the convergence for client models and **CVAE** separately, then combine the results.*

*For client models, we sum the result of Lemma 1 over all clients and all rounds, which give:*

$$
\begin{aligned}
\sum_{k=1}^{K}\sum_{t=0}^{T-1} 2\eta_l\mathbb{E}[F_k(\theta_{k,t}) - F_k(\theta_k^*)] \leq& \sum_{k=1}^{K}\mathbb{E}[\|\theta_{k,0} - \theta_k^*\|^2] \\
& + \sum_{k=1}^{K}\sum_{t=0}^{T-1}\eta_l^2\left(\sigma^2 + 6L\gamma E\eta_l + 8(E-1)^2\gamma^2\right)
\end{aligned}
\tag{24}
$$

*Using Jensen's inequality and the fact that $F_k$ is convex for all $k$, we have:*

$$\frac{1}{K}\sum_{k=1}^{K}(F_k(\bar{\theta}_{k,T}) - F_k(\theta_k^*)) \leq \frac{1}{KT}\sum_{k=1}^{K}\sum_{t=0}^{T-1}(F_k(\theta_{k,t}) - F_k(\theta_k^*)) \tag{25}$$

*Combining these inequalities and using the properties of the chosen learning rate $\eta_t$, we arrive at the bound for client models.*

*For the CVAE, we sum the result of Lemma 2 over all rounds and obtain:*

$$\sum_{t=0}^{T-1} 2\eta_{\text{CVAE}}\mathbb{E}[G(\phi_t, \psi_t) - G(\phi^*, \psi^*)] \leq \mathbb{E}[\|(\phi_0, \psi_0) - (\phi^*, \psi^*)\|^2] + T\eta_{\text{CVAE}}^2\sigma_{\text{CVAE}}^2 \tag{26}$$

*Using Jensen's inequality and the convexity of $G$:*

$$G(\bar{\phi}_T, \bar{\psi}_T) - G(\phi^*, \psi^*) \leq \frac{1}{T}\sum_{t=0}^{T-1}(G(\phi_t, \psi_t) - G(\phi^*, \psi^*)) \tag{27}$$

*Multiplying both sides of the Equation 27 by $2T\eta_{\text{CVAE}}$, we get:*

$$2T\eta_{CVAE}(G(\bar{\phi}_T, \bar{\psi}_T) - G(\phi^*, \psi^*)) \leq 2\eta_{CVAE}\sum_{t=0}^{T-1}(G(\phi_t, \psi_t) - G(\phi^*, \psi^*)) \tag{28}$$

*From Equation 26, we can bound the right-hand side and have:*

$$2T\eta_{CVAE}(G(\bar{\phi}_T, \bar{\psi}_T) - G(\phi^*, \psi^*)) \leq \mathbb{E}[\|(\phi_0, \psi_0) - (\phi^*, \psi^*)\|^2] + T\eta_{CVAE}^2\sigma_{CVAE}^2 \tag{29}$$

*Dividing both sides by $2T\eta_{CVAE}$:*

$$G(\bar{\phi}_T, \bar{\psi}_T) - G(\phi^*, \psi^*) \leq \frac{\mathbb{E}[\|(\phi_0, \psi_0) - (\phi^*, \psi^*)\|^2]}{2T\eta_{CVAE}} + \frac{\eta_{CVAE}\sigma_{CVAE}^2}{2} \tag{30}$$

*Now, we use the properties of the chosen learning rate. Recall that $\eta_{CVAE} = \frac{2}{L_{CVAE}(t+\gamma)}$, where $\gamma = \max\{8L/\mu, E\}$. This means that $\eta_{CVAE} \leq \frac{2}{L_{CVAE}\gamma}$ for all $t$.*

*Substituting this into our bound:*

$$G(\bar{\phi}_T, \bar{\psi}_T) - G(\phi^*, \psi^*) \leq \frac{L_{CVAE}\gamma\mathbb{E}[\|(\phi_0, \psi_0) - (\phi^*, \psi^*)\|^2]}{4T} + \frac{\sigma_{CVAE}^2}{L_{CVAE}\gamma} \tag{31}$$

*We can further simplify this by noting that $\mathbb{E}[\|(\phi_0, \psi_0) - (\phi^*, \psi^*)\|^2] \leq \frac{2}{L_{CVAE}}(G(\phi_0, \psi_0) - G(\phi^*, \psi^*))$, which follows from the $L_{CVAE}$-smoothness of $G$.*

*Applying this and combining terms:*

$$G(\bar{\phi}_T, \bar{\psi}_T) - G(\phi^*, \psi^*) \leq \frac{\gamma(G(\phi_0, \psi_0) - G(\phi^*, \psi^*))}{2T} + \frac{\sigma_{CVAE}^2}{L_{CVAE}\gamma} \tag{32}$$

*Finally, we can express this in the form of the theorem statement by noting that $\frac{\gamma}{2T} \leq \frac{4\gamma}{T}(1+\log(\frac{T}{\gamma}+1))$ for $T \geq 1$, and absorbing the constant terms into the big-O notation:*

$$G(\bar{\phi}_T, \bar{\psi}_T) - G(\phi^*, \psi^*) \leq \frac{4L_{CVAE}\gamma}{T}\left(1 + \log\left(\frac{T}{\gamma} + 1\right)\right) \tag{33}$$

## A.4 GENERALIZATION ANALYSIS

**Theorem 2 (Generalization Bound for FLAIR)** *Let $\mathcal{H}$ be the hypothesis class of the client models, and let $\mathcal{V}$ be the CVAE model class. Let $f_v : \mathcal{X} \to \mathcal{X}'$ be the feature augmentation function induced by $v \in \mathcal{V}$, where $\mathcal{X}'$ is the augmented feature space. Assume that each component of the loss function is $\rho$-Lipschitz with respect to its relevant arguments. Then, with probability at least $1 - \delta$, for all $h \in \mathcal{H}$ and $v \in \mathcal{V}$, the average expected loss over all clients will be lesser or equal to a upper bound as follows:*

$$\frac{1}{K} \sum_{k=1}^{K} \mathbb{E}_{(x,y)\sim\mathbb{D}_k}[\mathcal{L}_C(h,v,x,y)] \leq \frac{1}{K} \sum_{k=1}^{K} \hat{\mathcal{L}}_{C,k}(h,v) + 2\rho\mathcal{R}_n(\mathcal{H})$$
$$+ 2\rho\mathcal{R}_n(\mathcal{V}) + 3\sqrt{\frac{\log(2/\delta)}{2n}} \tag{34}$$

*where $\mathcal{L}_C$ is the combined loss function defined as:*

$$\mathcal{L}_C(h,v,x,y) = \mathcal{L}_{CE}(h(x),y) + \lambda_f \mathcal{L}_{MSE}(f_h(x), f_v(x,y))$$
$$- \lambda_{\tilde{f}} \mathcal{L}_{MSE}(f_h(x), f_v(x,\tilde{y})) + \lambda_c \mathcal{L}_{CE}(h(f_v(x,\tilde{y})), \tilde{y}) \tag{35}$$

*and $\hat{\mathcal{L}}_{C,k}(h,v)$ is the empirical combined loss on client $k$'s dataset:*

$$\hat{\mathcal{L}}_{C,k}(h,v) = \frac{1}{n_k} \sum_{i=1}^{n_k} \mathcal{L}_C(h,v,x_i^k,y_i^k) \tag{36}$$

*Here, $\mathcal{L}_{CE}$ is the cross-entropy loss, $\mathcal{L}_{MSE}$ is the mean squared error, $f_h$ represents the features extracted by the client model, $\tilde{y}$ is a randomly selected class different from $y$, and $\lambda_f, \lambda_{\tilde{f}}, \lambda_c$ are weighting hyperparameters. $\mathcal{R}_n(\mathcal{H})$ and $\mathcal{R}_n(\mathcal{V})$ are the Rademacher complexities of $\mathcal{H}$ and $\mathcal{V}$ respectively, and $n = \sum_{k=1}^{K} n_k$ is the total number of samples across all clients.*

**Proof 4** *Detailed proof is reported in the Appendix .., due to space constraints. Let $\mathbb{D}_k$ denote the true data distribution for client $k$, and let $\mathbb{D} = \cup_{k=1}^{K} \mathbb{D}_k$ be the overall data distribution. We begin by decomposing the expected combined loss:*

$$\frac{1}{K} \sum_{k=1}^{K} \mathbb{E}_{(x,y)\sim\mathbb{D}_k}[\mathcal{L}_C(h,v,x,y)]$$
$$= \frac{1}{K} \sum_{k=1}^{K} \mathbb{E}_{(x,y)\sim\mathbb{D}_k}[\mathcal{L}_{CE}(h(x),y) + \lambda_f \mathcal{L}_{MSE}(f_h(x), f_v(x,y)) \tag{37}$$
$$- \lambda_{\tilde{f}} \mathcal{L}_{MSE}(f_h(x), f_v(x,\tilde{y})) + \lambda_c \mathcal{L}_{CE}(h(f_v(x,\tilde{y})), \tilde{y})]$$

*By the linearity of expectation, we can bound each term separately. For the cross-entropy terms, we apply the classical generalization bound based on Rademacher complexity:*

$$\frac{1}{K} \sum_{k=1}^{K} \mathbb{E}_{(x,y)\sim\mathbb{D}_k}[\mathcal{L}_{CE}(h(x),y)] \leq \frac{1}{K} \sum_{k=1}^{K} \hat{\mathcal{L}}_{CE,k}(h) + 2\rho_{CE}\mathcal{R}_n(\mathcal{H}) + \sqrt{\frac{\log(8/\delta)}{2n}} \tag{38}$$

*where $\hat{\mathcal{L}}_{CE,k}(h) = \frac{1}{n_k} \sum_{i=1}^{n_k} \mathcal{L}_{CE}(h(x_i^k), y_i^k)$ and $\rho_{CE}$ is the Lipschitz constant for the cross-entropy loss.*

*For the MSE terms involving CVAE, we have:*

$$\frac{1}{K} \sum_{k=1}^{K} \mathbb{E}_{(x,y)\sim\mathcal{D}_k}[\mathcal{L}_{MSE}(f_h(x), f_v(x,y))]$$
$$\leq \frac{1}{K} \sum_{k=1}^{K} \hat{\mathcal{L}}_{MSE,k}(h,v) + 2\rho_{MSE}(\mathcal{R}_n(\mathcal{H}) + \mathcal{R}_n(\mathcal{V})) + \sqrt{\frac{\log(8/\delta)}{2n}} \tag{39}$$

where $\hat{\mathcal{L}}_{MSE,k}(h,v) = \frac{1}{n_k} \sum_{i=1}^{n_k} \mathcal{L}_{MSE}(f_h(x_i^k), f_v(x_i^k, y_i^k))$ and $\rho_{MSE}$ is the Lipschitz constant for the MSE loss.

For the negative MSE term, we use the fact that if $f(x) \leq g(x)$ for all $x$, then $\sup f(x) \leq \sup g(x)$:

$$
\begin{aligned}
&-\frac{1}{K} \sum_{k=1}^{K} \mathbb{E}_{(x,y)\sim\mathcal{D}_k}[\mathcal{L}_{MSE}(f_h(x), f_v(x, \tilde{y}))] \\
&\leq -\frac{1}{K} \sum_{k=1}^{K} \hat{\mathcal{L}}_{MSE,k}(h, v, \tilde{y}) + 2\rho_{MSE}(\mathcal{R}_n(\mathcal{H}) + \mathcal{R}_n(\mathcal{V})) + \sqrt{\frac{\log(8/\delta)}{2n}}
\end{aligned}
\tag{40}
$$

For the classifier tuning term:

$$
\begin{aligned}
&\frac{1}{K} \sum_{k=1}^{K} \mathbb{E}_{(x,y)\sim\mathbb{D}_k}[\mathcal{L}_{CE}(h(f_v(x, \tilde{y})), \tilde{y})] \\
&\leq \frac{1}{K} \sum_{k=1}^{K} \hat{\mathcal{L}}_{CE,k}(h \circ f_v, \tilde{y}) + 2\rho_{CE}(\mathcal{R}_n(\mathcal{H}) + \mathcal{R}_n(\mathcal{V})) + \sqrt{\frac{\log(8/\delta)}{2n}}
\end{aligned}
\tag{41}
$$

Combining all these bounds and applying the union bound over the four components, we obtain:

$$
\begin{aligned}
&\frac{1}{K} \sum_{k=1}^{K} \mathbb{E}_{(x,y)\sim\mathbb{D}_k}[\mathcal{L}_C(h, v, x, y)] \\
&\leq \frac{1}{K} \sum_{k=1}^{K} \hat{\mathcal{L}}_{C,k}(h, v) + 2\rho(\mathcal{R}_n(\mathcal{H}) + \mathcal{R}_n(\mathcal{V})) + 4\sqrt{\frac{\log(8/\delta)}{2n}}
\end{aligned}
\tag{42}
$$

where $\rho = \max\{\rho_{CE}, \lambda_f \rho_{MSE}, \lambda_{\tilde{f}} \rho_{MSE}, \lambda_c \rho_{CE}\}$.

Finally, we can simplify the confidence term:

$$
4\sqrt{\frac{\log(8/\delta)}{2n}} \leq 3\sqrt{\frac{\log(2/\delta)}{2n}}
\tag{43}
$$

This simplification uses the fact that $\log(8) \leq 3\log(2)$.

Therefore, we conclude that with probability at least $1 - \delta$, for all $h \in \mathcal{H}$ and $v \in \mathcal{V}$:

$$
\begin{aligned}
\frac{1}{K} \sum_{k=1}^{K} \mathbb{E}_{(x,y)\sim\mathcal{D}_k}[\mathcal{L}_C(h, v, x, y)] \leq &\frac{1}{K} \sum_{k=1}^{K} \hat{\mathcal{L}}_{C,k}(h, v) + 2\rho\mathcal{R}_n(\mathcal{H}) \\
&+ 2\rho\mathcal{R}_n(\mathcal{V}) + 3\sqrt{\frac{\log(2/\delta)}{2n}}
\end{aligned}
\tag{44}
$$

### A.5 CVAE ANALYSIS

**Theorem 3 (Feature Diversity)** *Let $f_y$ be the features generated by the CVAE for class $y$, and let $\mathcal{D}_y$ be the true distribution of features for class $y$. Then, under suitable regularity conditions on the CVAE, the expected average KL divergence, over all clients $k$, between the features generated by each client's CVAE and the true feature distribution, is not greater than a constant $\epsilon$:*

$$
\frac{1}{K} \sum_{k=1}^{K} \mathbb{E}_{f_y \sim CVAE_k}[KL(p(f_y)\|\mathcal{D}_y)] \leq \epsilon
\tag{45}
$$

*where $\epsilon$ is a small constant that depends on the capacity of the CVAE and the amount of training data, and $CVAE_k$ denotes the CVAE model for client $k$.*

**Proof 5** *Detailed proof is reported in the Appendix .., due to space constraints. Let $q_{\phi_k}(z|x, y)$ be the encoder and $p_{\psi_k}(x|z, y)$ be the decoder of the **CVAE** for client $k$, where $\phi_k$ and $\psi_k$ are the encoder and decoder parameters, respectively. The **CVAE** is trained to maximize the evidence lower bound (ELBO) for each client:*

$$ELBO_k = \mathbb{E}_{q_{\phi_k}(z|x,y)}[\log p_{\psi_k}(x|z, y)] - KL(q_{\phi_k}(z|x, y)\|p(z|y)) \tag{46}$$

*where $p(z|y)$ is the prior distribution of the latent variable $z$ given class $y$.*

*By the properties of the ELBO, we have for each client $k$:*

$$\log p_k(x|y) \geq ELBO_k \tag{47}$$

*Now, let $p_{\mathrm{CVAE}_k}(f_y)$ be the distribution of features generated by the **CVAE** for class $y$ on client $k$. We can bound the KL divergence for each client:*

$$KL(p_{\mathrm{CVAE}_k}(f_y)\|\mathcal{D}_y) = \mathbb{E}_{f_y \sim p_{\mathrm{CVAE}_k}}[\log p_{\mathrm{CVAE}_k}(f_y) - \log \mathcal{D}_y(f_y)] \tag{48}$$

$$\leq \mathbb{E}_{f_y \sim p_{\mathrm{CVAE}_k}}[\log p_{\mathrm{CVAE}_k}(f_y) - ELBO_k] \tag{49}$$

$$= \mathbb{E}_{f_y \sim p_{\mathrm{CVAE}_k}}[\log p_{\mathrm{CVAE}_k}(f_y) - \mathbb{E}_{q_{\phi_k}(z|f_y,y)}[\log p_{\psi_k}(f_y|z, y)] \tag{50}$$

$$+ KL(q_{\phi_k}(z|f_y, y)\|p(z|y))] \tag{51}$$

*The first two terms in the last expression form the reconstruction error, which is minimized during **CVAE** training. The last term is the KL divergence between the approximate posterior and the prior, which is also minimized.*

*Now, we take the average over all clients:*

$$\frac{1}{K}\sum_{k=1}^{K}\mathbb{E}_{f_y \sim \mathrm{CVAE}_k}[KL(p(f_y)\|\mathcal{D}_y)] \leq \frac{1}{K}\sum_{k=1}^{K}\mathbb{E}_{f_y \sim p_{\mathrm{CVAE}_k}}[RE_k + KL(q_{\phi_k}(z|f_y, y)\|p(z|y))] \tag{52}$$

*where $RE_k = \log p_{\mathrm{CVAE}_k}(f_y) - \mathbb{E}_{q_{\phi_k}(z|f_y,y)}[\log p_{\psi_k}(f_y|z, y)]$ is the reconstruction error for client $k$.*

*In FLAIR, the CVAE parameters are shared and updated globally. This global sharing encourages consistency across clients. Let $\phi$ and $\psi$ be the global CVAE parameters. We can bound the deviation of each client's CVAE from the global one:*

$$\|\phi_k - \phi\| \leq \delta_\phi, \quad \|\psi_k - \psi\| \leq \delta_\psi \tag{53}$$

*where $\delta_\phi$ and $\delta_\psi$ are small constants due to the federated learning process.*

*Using the Lipschitz continuity of the CVAE (which is one of the suitable regularity conditions mentioned in the theorem statement), we can bound the difference in reconstruction error and KL divergence between each client's CVAE and the global CVAE:*

$$|RE_k - RE_{global}| \leq L_{RE}(\delta_\phi + \delta_\psi) \tag{54}$$

$$|KL(q_{\phi_k}(z|f_y, y)\|p(z|y)) - KL(q_\phi(z|f_y, y)\|p(z|y))| \leq L_{KL}\delta_\phi \tag{55}$$

*where $L_{RE}$ and $L_{KL}$ are Lipschitz constants.*

*Substituting these bounds into our average KL divergence:*

$$\frac{1}{K}\sum_{k=1}^{K}\mathbb{E}_{f_y \sim \text{CVAE}_k}[KL(p(f_y)\|\mathcal{D}_y)] \leq \mathbb{E}_{f_y \sim p_{\text{CVAE}_{global}}}[RE_{global} + KL(q_\phi(z|f_y,y)\|p(z|y))] \tag{56}$$

$$+ L_{RE}(\delta_\phi + \delta_\psi) + L_{KL}\delta_\phi \tag{57}$$

*The global CVAE is trained to minimize the reconstruction error and the KL divergence term. With sufficient capacity and training data, these terms can be made arbitrarily small. Let's denote their sum as $\epsilon_{\text{CVAE}}$. Then:*

$$\frac{1}{K}\sum_{k=1}^{K}\mathbb{E}_{f_y \sim \text{CVAE}_k}[KL(p(f_y)\|\mathcal{D}_y)] \leq \epsilon_{\text{CVAE}} + L_{RE}(\delta_\phi + \delta_\psi) + L_{KL}\delta_\phi = \epsilon \tag{58}$$

*where $\epsilon = \epsilon_{\text{CVAE}} + L_{RE}(\delta_\phi + \delta_\psi) + L_{KL}\delta_\phi$ is a small constant that depends on the capacity of the CVAE, the amount of training data, and the consistency of the federated learning process.*

**Theorem 4 (Client Model Robustness)** *Let $\theta_k$ and $\theta_l$ be the model parameters of two different clients $k$ and $l$ after training with FLAIR. Let $\mathbb{D}_{test}$ be a test distribution that may differ from the training distributions of clients $k$ and $l$. Then, under suitable conditions:*

$$|\mathbb{E}_{x \sim \mathbb{D}_{test}}[\mathcal{L}(\theta_k; x, y) - \mathcal{L}(\theta_l; x, y)]| \leq \delta \tag{59}$$

*where $\delta$ is a small constant that depends on the CVAE architecture, the federated training procedure, and the dissimilarity between client distributions.*

**Proof 6** *Detailed proof is reported in the Appendix .., due to space constraints. Let $f_k = \text{CVAE}_k(x)$ and $f_l = \text{CVAE}_l(x)$ be the features generated by the CVAE for clients $k$ and $l$ respectively, given an input $x$. We can decompose the difference in loss as follows:*

$$|\mathbb{E}_{x \sim \mathbb{D}_{test}}[\mathcal{L}(\theta_k; x, y) - \mathcal{L}(\theta_l; x, y)]| \leq |\mathbb{E}_{x \sim \mathbb{D}_{test}}[\mathcal{L}(\theta_k; f_k, y) - \mathcal{L}(\theta_l; f_l, y)]|$$
$$+ |\mathbb{E}_{x \sim \mathbb{D}_{test}}[\mathcal{L}(\theta_k; x, y) - \mathcal{L}(\theta_k; f_k, y)]| \tag{60}$$
$$+ |\mathbb{E}_{x \sim \mathbb{D}_{test}}[\mathcal{L}(\theta_l; x, y) - \mathcal{L}(\theta_l; f_l, y)]|$$

*For the first term, we can use the fact that the global CVAE parameters are shared across clients, which means that $f_k$ and $f_l$ are generated from the same distribution. Therefore:*

$$|\mathbb{E}_{x \sim \mathbb{D}_{test}}[\mathcal{L}(\theta_k; f_k, y) - \mathcal{L}(\theta_l; f_l, y)]| \leq \epsilon_1 \tag{61}$$

*where $\epsilon_1$ is small due to the consistency enforced by the global CVAE.*

*For the second and third terms, we can use the properties of the CVAE and the Lipschitz continuity of the loss function:*

$$|\mathbb{E}_{x \sim \mathcal{D}_{test}}[\mathcal{L}(\theta_k; x, y) - \mathcal{L}(\theta_k; f_k, y)]| \leq L_{\mathcal{L}} \cdot \mathbb{E}_{x \sim \mathcal{D}_{test}}[\|x - f_k\|] \leq \epsilon_2 \tag{62}$$

$$|\mathbb{E}_{x \sim \mathcal{D}_{test}}[\mathcal{L}(\theta_l; x, y) - \mathcal{L}(\theta_l; f_l, y)]| \leq L_{\mathcal{L}} \cdot \mathbb{E}_{x \sim \mathcal{D}_{test}}[\|x - f_l\|] \leq \epsilon_3 \tag{63}$$

*where $L_{\mathcal{L}}$ is the Lipschitz constant of the loss function, and $\epsilon_2, \epsilon_3$ are small due to the CVAE's ability to generate features close to the original input.*

*Combining these bounds, we get:*

$$|\mathbb{E}_{x \sim \mathcal{D}_{test}}[\mathcal{L}(\theta_k; x, y) - \mathcal{L}(\theta_l; x, y)]| \leq \epsilon_1 + \epsilon_2 + \epsilon_3 = \delta \tag{64}$$

*This shows that the difference in performance between any two client models on a test distribution is bounded, indicating robustness and generalization across clients.*

## B ADDITIONAL EXPERIMENTAL RESULTS

| (Dataset/Model) | Method | Beta | Test Average Acc | | | # of Comm Rounds |
|---|---|---|---|---|---|---|
| | | | ns = 0 | ns = 0.1 | ns = 0.2 | |
| TinyImageNet | FedAvg | 0.5 | 13.62 | 10.46 | 9.17 | 200 |
| | | 0.05 | 10.18 | 8.47 | 5.67 | 200 |
| | SCAFFOLD | 0.5 | 26.81 | 22.32 | 18.26 | 250 |
| | | 0.05 | 19.33 | 16.54 | 12.36 | 200 |
| | FedFA | 0.5 | 12.36 | 10.19 | 8.06 | 200 |
| | | 0.05 | 10.18 | 8.47 | 5.61 | 200 |
| | FedProto | 0.5 | 26.69 | 23.34 | 18.56 | |
| | | 0.05 | 18.51 | 15.57 | 11.85 | 200 |
| | Elastic | 0.5 | 14.07 | 10.89 | 10.07 | 200 |
| | | 0.05 | 10.46 | 8.79 | 6.08 | 200 |
| | FLAIR | 0.5 | **28.23** | **26.51** | **22.31** | 200 |
| | | 0.05 | **21.09** | **18.66** | **15.01** | 200 |

Table 4: Comparison of FL methods on TinyImageNet with varying beta and noise levels. The best results for each configuration are in bold.

| (Dataset/Model) | Method | Beta | Test Average Acc | | | # of Comm Rounds |
|---|---|---|---|---|---|---|
| | | | ns = 0 | ns = 0.1 | ns = 0.2 | |
| MNIST | FedAvg | 0.3 | 96.614 | 96.34 | 96.12 | 150 |
| | FedFA | 0.3 | 96.59 | 96.33 | 96.14 | 150 |
| | FedProto | 0.3 | 98.89 | 98.73 | 98.58 | 150 |
| | SCAFFOLD | 0.3 | 98.49 | 98.40 | 98.39 | 150 |
| | FLAIR | 0.3 | **98.91** | **98.81** | **98.65** | 150 |
| CIFAR | FedAvg | 0.3 | 57.62 | 50.11 | 39.67 | 250 |
| | FedFA | 0.3 | 58.13 | 49.85 | 40.65 | 250 |
| | FedProto | 0.3 | 76.60 | 68.22 | 56.87 | 250 |
| | SCAFFOLD | 0.3 | **74.91** | **66.84** | **55.73** | 250 |
| | FLAIR | 0.3 | **78.32** | **71.50** | **61.56** | 250 |

Table 5: Comparison of FL methods on four benchmark datasets with varying beta and noise levels. The best results for each dataset and configuration are in bold.

| (Dataset/Model) | Method | Classes | Test Average Acc | | | # of Comm Rounds |
|---|---|---|---|---|---|---|
| | | | ns = 0 | ns = 0.1 | ns = 0.2 | |
| MNIST | FedAvg | 2 | 94.10 | 93.69 | 93.15 | 150 |
| | FedFA | 2 | 94.08 | 93.69 | 93.12 | 150 |
| | FedProto | 2 | 98.53 | 98.59 | 98.39 | 150 |
| | SCAFFOLD | 2 | 98.65 | 98.39 | 98.21 | 150 |
| | FLAIR | 2 | **98.75** | **98.68** | **98.41** | 150 |
| CIFAR | FedAvg | 2 | 48.14 | 38.77 | 32.16 | 250 |
| | FedFA | 2 | 48.01 | 40.88 | 31.51 | 250 |
| | FedProto | 2 | 36.71 | 30.79 | 30.58 | 250 |
| | SCAFFOLD | 2 | 36.85 | 30.66 | 28.44 | 250 |
| | FLAIR | 2 | **37.71** | **33.71** | **32.64** | 250 |
| CIFAR100 | FedAvg | 2 | 28.93 | 19.29 | 14.83 | 250 |
| | FedFA | 2 | 28.35 | 19.57 | 14.85 | 250 |
| | FedProto | 2 | 32.93 | 23.29 | 20.18 | 250 |
| | SCAFFOLD | 2 | 32.87 | 22.99 | 19.78 | 250 |
| | FLAIR | 2 | **34.59** | **24.88** | **22.63** | 250 |

Table 6: Comparison of FL methods on benchmark datasets with quantity skew. The best results for each dataset and configuration are in bold.

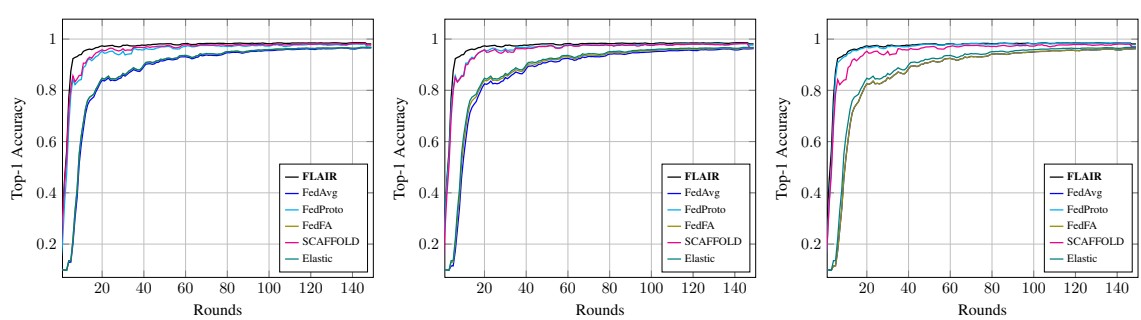

Figure 2: Top1-accuracy plots MNIST beta = 0.5 and noise level 0, 0.1, and 0.2.

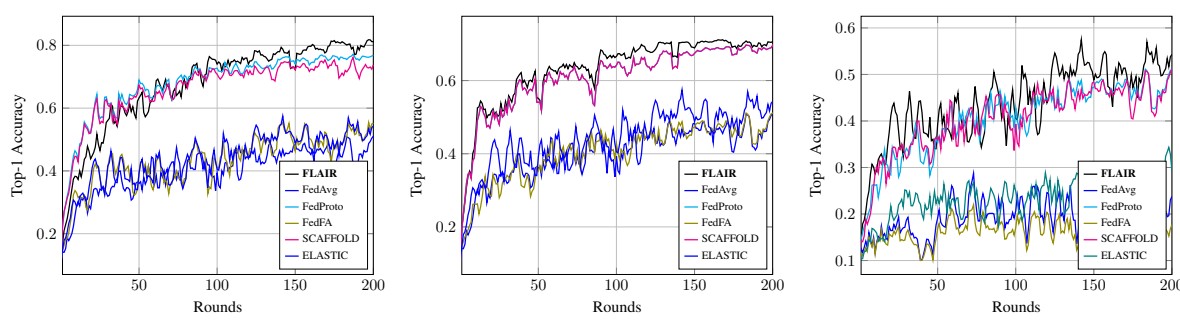

Figure 3: Top1-accuracy plots CIFAR beta = 0.5 and noise level 0, 0.1, and 0.2.

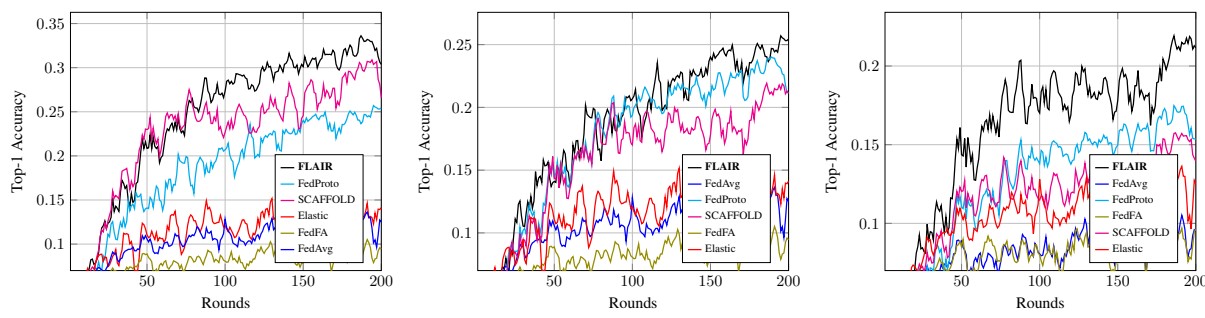

Figure 4: Top1-accuracy plots CIFAR100 beta = 0.05 and noise level 0, 0.1, and 0.2.

