# OpenReview forum: "FLAIR: FEDERATED LEARNING WITH AUGMENTED AND IMPROVED FEATURE REPRESENTATIONS"
_ICLR.cc/2025/Conference — ICLR 2025 Conference Withdrawn Submission_

### Official Review · Reviewer_aHcL · 2024-10-25

**Soundness:** 2
**Presentation:** 1
**Contribution:** 2
**Rating:** 3
**Confidence:** 4

**Summary:**

This paper introduces a method, FLAIR, to address statistical heterogeneity among client datasets in the global federated learning task, specifically targeting communication overhead and privacy concerns associated with methods that exchange additional information. The authors tackle this problem by using an alternative approach to knowledge sharing through a CVAE model integrated into both server and client. To incorporate the CVAE into the local model training process, they modify the loss function to train the local model with three distinct components. Additionally, they introduce a Reptile meta-learning-based procedure to train the CVAE model. Extensive experiments are conducted across various scenarios to validate the effectiveness of the proposed method. Overall, the authors present a promising model to address the limitations of additional information-sharing methods, and the experimental results suggest its effectiveness. However, a more detailed analysis of the CVAE's role in performance improvements is necessary.

**Strengths:**

1. The authors propose an alternative approach to transmitting local model parameters directly, aiming to enhance both communication efficiency and privacy while improving overall performance.

2. In addition to experiments measuring accuracy, the authors also conduct experiments to quantify the level of privacy, providing numerical evidence of the proposed algorithm's privacy guarantees.

**Weaknesses:**

1. The Conditional Variational Autoencoder (CVAE) applied in the paper was introduced in 2015, and since then, various other VAE techniques have been proposed. Therefore, alternative VAE techniques could potentially serve as mechanisms for sharing information between the server and clients. However, the paper does not clearly explain why CVAE, in particular, was chosen for the federated learning framework.

2. This lack of clarity is also reflected in the experimental results. The experiments do not include an ablation study that would clarify the specific role of CVAE in the proposed method. For instance, an ablation could involve adding each of the three individual losses in the local model's loss function one by one or replacing CVAE with a vanilla VAE to observe the impact on performance.

3. Furthermore, it is unclear which part of the algorithm is dedicated to the “Reptile-based” approach. The algorithm seems to resemble a standard CVAE training procedure, so clarification is needed on the difference between the vanilla update and the Reptile-based update in this context.

**Questions:**

1. Is it "Class VAE" or "Conditional VAE"? The abbreviation “CVAE” is mentioned in both the abstract and introduction but appears to refer to different terms.

2. In Table 1, what is the difference between the total number of clients and the number of local models? If each client has a single local model, shouldn't the FLAIR method also be computed as 𝑂(𝐸×𝑆_𝑡)?

3. What is the computational overhead of the additional update step for CVAE after local model training?

4. In Section 2.2, what is the purpose of generating and including 𝑦_tild in training? Unlike other losses, the reason for including this loss is not explained explicitly.

5. In Section 2.5 on line 307, where are lines 34 and 35?

6. In Section 4.2, line 356, the phrase “three widely-used datasets” should be revised to “four widely-used datasets” since the experiments are conducted on MNIST, CIFAR-10, CIFAR-100, and TinyImagenet.

7. For reproducibility, it would be helpful to specify the hardware environment used in the experiments, including details like OS, CPU, and GPU.

---

### Official Review · Reviewer_UNqK · 2024-11-03

**Soundness:** 2
**Presentation:** 2
**Contribution:** 2
**Rating:** 3
**Confidence:** 5

**Summary:**

This paper proposes a novel FL method called FLAIR, which aims to strike a balance between communication efficiency, privacy protection, and adaptability to heterogeneous data distributions. Specifically, FLAIR utilizes Class Variational Autoencoders (CVAE) for feature augmentation, mitigating class imbalance and missing class issues. It also incorporates Reptile meta-training to facilitate knowledge transfer between model updates, adapting to dynamic feature shifts. To generalize model update, FLAIR shares only local CVAE parameters instead of local model parameters, which reduces both communication costs and privacy risks. Empirical experiments with extensive analysis on image classification datasets demonstrate the superiority of FLAIR in terms of test accuracy.

**Strengths:**

1. The paper is well-written and easy to follow.
2. Data-free black-box knowledge transfer across heterogeneous clients in Federated Learning (FL) is interesting and promising.

**Weaknesses:**

1. There are some vague and confusing expressions in the paper, which seriously reduces the readability of the paper.

2. This study lacks innovation, as previous studies have used similar simpler strategies but have a wider range of applicability, such as FL for model heterogeneity.

3. Lack of ablation research on multiple hyperparameters in the proposed method.

**Questions:**

1. The display of FLAIR in Figure 1 is complex and has low readability. Merely relying on Figure 1 cannot effectively grasp the contribution of this work. Suggest further optimizing Figure 1 to make it more readable.

2. There are many details errors here, such as the fact that the parameter $\theta$ in Eq. (2) does not indicate whose parameter it is, and is $\theta_{k, t} $(see Eq. (2)) the same as $\theta_{t, k} $(see Eq. (3))? I strongly recommend the author to double check the details of the wording in the paper.

3. Existing work [1] seems to use a similar strategy (using variational autoencoder). However, this paper lacks attention and comparison to it. I think it should be an important comparison method.

4. I did not see the setting of the number of clients during the experiment, that is, the value of $K$.

5.  All the report results in the paper are the final evaluation indicators, such as accuracy, which is insufficient. Therefore, the learning curves and communication rounds should also be reported to demonstrate the training process of FLAIR and baselines.

6. From the method description section, it can be inferred that multiple hyperparameters such as $\lambda_f$, $\lambda_\widetilde{f}$, $\lambda_c$ and $\lambda$ are introduced during the training process of FLAIR. However, the ablation experiment lacks detailed numerical experimental research on them.

7. I don't understand the significance of the content reported in Table 3. Since FLAIR cannot obtain privacy estimates in MIA, Gradient Leave, and Info Theoretic, why report them?

[1] Heinbaugh C E, Luz-Ricca E, Shao H. Data-free one-shot federated learning under very high statistical heterogeneity[C]//The Eleventh International Conference on Learning Representations. 2023.

---

### Official Review · Reviewer_xtHD · 2024-11-03

**Soundness:** 2
**Presentation:** 2
**Contribution:** 3
**Rating:** 3
**Confidence:** 5

**Summary:**

"FLAIR: Federated Learning with Augmented and Improved Feature Representations" introduces the use of Class Variational Autoencoders (CVAE) to tackle challenges in federated learning, specifically addressing non-IID data distributions and communication overhead. The authors innovate by leveraging autoencoders to reduce communication costs, pointing out that traditional gradient sharing increases overhead. This new method is evaluated against different datasets and baseline methods as a solution to train models under non-iid setting. The authors also present theoretical guarantees of convergence, generalizability and robustness.

**Strengths:**

-   **Addressing Non-IID Data and Communication Constraints:**  The authors  address  training models with non-IID data distributions while facing communication overhead limitations by proposing the use of Class Variational Autoencoders (CVAE).


-   **Identification of  Communication Overhead in Federated Learning:**  The paper points out that gradient sharing in standard Federated Learning algorithms leads to increased communication overhead and how FLAIR attempts to address this.

-   **Multiple Experiments:**  Although the explicability of these experiments is a main concern which I explain in the weakness section, the study does evaluate the proposed approach against five baseline methods across three different datasets.

-   The authors provide the code, facilitating reproducibility of their experiments and results.

**Weaknesses:**

**Claim 1 is not substantiated:**
The **central** claim of the paper is that communication complexity is reduced, as indicated in Table 1. We know that FLAIR achieves  O(E) communication complexity, but it is not evident how this is lower than  O(St) or other methods ( ie: no clear evidence that E << St ). Additionally, assuming E << St , while the choice to exchange CVAE parameters instead of model gradients appears to be a valid solution for reducing per-round complexity, especially since the baselines considered all use model gradients, this does raise the question: Are the authors the first to propose this alternative in distributed learning? If so, this should be prominently highlighted. If not, why were other baselines that do not exchange model gradients not considered?

**Claim 2 is not substantiated:**
FLAIR is claimed to reduce privacy risks. In my opinion, this claim is unsubstantiated. The authors use the term "privacy" very loosely and mention terms like "privacy attacks" (Line 171) without clear definitions. Even setting aside the lack of rigor in terminology, I require more convincing evidence regarding the types of attacks considered. Gradient leaks are not applicable since there is no gradient sharing, but **how does the algorithm protect against Membership Inference Attacks (MIA)?** This would imply that an adversarial server cannot distinguish between two datasets with a missing sample based on the exchanged CVAE parameters, similar to standard Differential Privacy (DP) settings. Maybe a different definition of MIA is considered. Regardless, no details are provided on how exactly the privacy metrics are evaluated in this context.

**Multiple Unsubstantiated Claims eg:**

-   **Line 429:**  "Performance gap widening as dataset complexity and heterogeneity increases." *What evidence supports this statement?*
-   **Line 445:**  "... noticeable jumps." *Where can I find the evidence for this claim?*

**Experimental Issues:**

-   **Beta Values:**  These are not clearly explained. There appears to be some relation in the label skew section, but it is unclear how this causes non-overlapping classes across different clients.
-   **Feature Skew Condition:**  This is not well-explained in my opinion, and there seem to be missing citations for these definitions, which exist in the literature but  not cited by the authors.
-   **Increasing Noise Levels Experiment:**  It is unclear whether this refers to DP-SGD. The paper does not explain this aspect adequately to me.
-   **Ablation Studies:**  There is no ablation on the choice of hyperparameters used in the combined loss function.

**Missing References/Grammatical Errors:**

-   **Line 169:**  Wording error
-  **Appendix**: What are proofs 4, 5, and 6, and how do they relate to the main theorems?

**Theoretical Concerns:**

-   **Theorem 1:**  Although the proof relies on strong convexity, my primary concern is how convergence guarantees **are provided despite client drift**. I did not replicate the proof, but it is surprising that unbounded client drift would still allow the model to converge. If this is indeed the case, the authors must highlight it.
-   **Theorem 2:**  It is unexpected that the generalization bound has no constraints on the choice of hyperparameters in the combined loss function.
-   **Theorems 3 and 4:**  These rely on unspecified regularity conditions. The authors mention "under suitable conditions" but do not provide these conditions.

**Models and Datasets:**
The study focuses on CNNs and vision tasks, which is not a major concern. However, a brief explanation of why only vision tasks were chosen would be beneficial.

**Questions:**

Refer to the Weaknesses Section

---

### Official Review · Reviewer_Dw88 · 2024-11-11

**Soundness:** 3
**Presentation:** 2
**Contribution:** 2
**Rating:** 3
**Confidence:** 3

**Summary:**

The authors of this work propose an FL framework that exploits conditional variational auto-encoders (CVAE) to tackle the data heterogeneity among clients participating in the training. They use a collaboratively learned CVAE to mitigate the heterogeneity that might arise from missing classes or extreme data distribution shifts. The authors provide theoretical analysis and experiments with their proposed solution

**Strengths:**

The experiment section of the work shows considerable improvements that FLAIR can achieve over prior work, especially in more extreme non-IID scenarios. The authors mix two key methods in machine learning to develop their solution (i) Reptile meta-learning, and (ii) CVAE.

**Weaknesses:**

The main weaknesses of the work include the following:
1. How does the work on CVAE compare against other prior works that explore VAE in FL?
2. Is it a fair evaluation of this work to say that instead of encoding and decoding the raw input and label the authors instead use an encoded version (which is compressed in terms of dimensionality)? Given this difference in input dimensionality, could the authors comment on the following:
	- are the number of model parameters between all the compared models similar? For example, in the case of FLAIR, the number of training parameters includes those from CVAE, the feature extractor, and the classifier.
	- What is the effect of the Reptile meta-learning algorithm on CVAE training? Have the authors seen any significant drop in performance if they do not use it? Also, from algorithm 1, it is unclear which steps represent the change based on the meta-learning.
	- What is the computational complexity of FLAIR? How do the baselines compare to FLAIR regarding FLOPS vs accuracy or wall-clock time vs accuracy?
3. Figure 1 is very hard to read. Authors should consider adding a legend to explain the chronology of steps, and the meaning of different arrows (dashed, solid, colored, etc.)
4. In terms of organization, authors could possibly present CVAE, Reptile meta-learning, etc. as preliminaries before jumping into the main training in 2.1 and 2.2. Also, the theorems section does not present any equations making it hard to understand the theoretical contribution of this work.
5. The role of equation 3 is not particularly clear in the text. The fact that ${\mathcal{L}}_{vf}$ and $\tilde{\mathcal{L}}_{vf}$ have opposing signs makes the loss similar to a min-max optimization. Is that the case? If so, the authors should clarify this point and present more details about the training objective. E.g.,
	- what is the impact of removing the inter-class and intra-class consistency losses?
	- How do the authors choose these $\lambda$ hyperparameters? How sensitive is the training to these?

**Questions:**

1. It is hard to visualize the heterogeneity based on the $\beta$ values that the authors report. Could the authors instead present some metrics about how the labels are distributed across clients in the appendix instead?
2. Could the authors highlight details of the training, such as the total number of clients and the number of clients sampled per training round? Similarly, does the number of local epochs denote those of the client model (feature encoder and classifier) or CVAE?
	- Also what is the effect of changing these? Does FLAIR scale well over different settings of client configurations such as low client participation?
3. Why do the representations from different CVAEs on different clients not clash with each other? Specifically, what do the authors think makes the representation space not overlap for different classes on different clients, say the space occupied by class 1 on client 1 is the same as class 2 on client 2?
4. In Algorithm 2:
	- during the initial phase, what representations do the CVAE train on? At this stage, the model parameters for feature extraction should not be well-trained.
5. Do the authors repeat their experiments over multiple trials? If so, do the numbers in the table represent the means of the trials? It would be helpful to look at the mean and standard deviations to understand the stability of the method
6. In Table 2 and Table 6, why do the authors consistently highlight only their performance numbers even if the baselines outperform them? It makes it quite hard to read the tables
7. Authors mention that "FLAIR exhibits faster convergence rates". Could they point to the section that shows these experiments?

---

### Note · Authors · 2024-11-25

I have read and agree with the venue's withdrawal policy on behalf of myself and my co-authors.